# Simulations of a protein fold switch reveal crowding-induced population shifts driven by disordered regions

Saman Bazmi [1,2], Bahman Seifi [1,2] & Stefan Wallin [1✉]

Macromolecular crowding effects on globular proteins, which usually adopt a single stable fold, have been widely studied. However, little is known about crowding effects on fold-switching proteins, which reversibly switch between distinct folds. Here we study the mutationally driven switch between the folds of $G_A$ and $G_B$, the two 56-amino acid binding domains of protein G, using a structure-based dual-basin model. We show that, in the absence of crowders, the fold populations $P_A$ and $P_B$ can be controlled by the strengths of contacts in the two folds, $\kappa_A$ and $\kappa_B$. A population balance, $P_A \approx P_B$, is obtained for $\kappa_B/\kappa_A = 0.92$. The resulting model protein is subject to crowding at different packing fractions, $\phi_c$. We find that crowding increases the $G_B$ population and reduces the $G_A$ population, reaching $P_B/P_A \approx 4$ at $\phi_c = 0.44$. We analyze the $\phi_c$-dependence of the crowding-induced $G_A$-to-$G_B$ switch using scaled particle theory, which provides a qualitative, but not quantitative, fit of our data, suggesting effects beyond a spherical description of the folds. We show that the terminal regions of the protein chain, which are intrinsically disordered only in $G_A$, play a dominant role in the response of the fold switch to crowding effects.

[1] Department of Physics and Physical Oceanography, Memorial University of Newfoundland, St. John's, NL A1B 3X7, Canada. [2] These authors contributed equally: Saman Bazmi, Bahman Seifi. ✉email: swallin@mun.ca

Most globular proteins rely on a single fold to carry out their function. However, recently proteins have been discovered with an ability to switch between different folds[1–4], a phenomenon called fold switching. By adopting an alternative structure, these fold-switching proteins (also termed metamorphic[5] or transformer[6] proteins) gain the ability to carry out an additional unrelated function. For example, a switch from a helical hairpin to a β-barrel transforms the *Escherichia coli* protein RfaH from a transcription factor to a translational activator[7]. Consistent with this view, fold switching is often regulated[8]. A range of cellular signals has been associated with fold switching, such as changes in salt concentration[9], redox conditions[10], and oligomerization[11]. Fold switching also underpins evolutionary changes in protein structure[12–14], in which case fold switching is driven by mutations.

In this work, we investigate the effects of macromolecular crowding on fold switching. To this end, we focus on the binding domains of protein G, $G_A$ and $G_B$, which form one of the most well-characterized fold switch systems[15] (see Fig. 1a). It was demonstrated that a set of substitution mutations can be found which drastically increases the sequence identity of $G_A$ and $G_B$, while still retaining their respective native structures and binding partners[15]. For example, the variants $G_A95$ and $G_B95$ differ in only 3 amino acid positions. Hence, three additional substitutions (L20A, I30F and L45Y) applied to $G_A95$ cause an abrupt switch from the 3α fold of $G_A$ to the $4β + α$ fold of $G_B$. Later it was shown that there are multiple ways in which a single substitution can tip the balance from one fold to the other, e.g., L20A applied to the variant $G_B98$-T25I[16]. These experiments on $G_A$ and $G_B$ were, however, carried out in dilute protein solutions and therefore in the absence of any crowding effects.

We carry out our simulations with a coarse-grained structure-based model, which we develop and test on the $G_A/G_B$ fold switch in the absence of crowders (see "Methods"). The structure-based approach involves constructing a potential energy landscape with a single basin of attraction by making native contacts attractive and non-native contacts repulsive. This type of modeling has provided important insights into several aspects of protein folding[17–20]. The natural extension to fold switching is a potential with dual basins of attractions corresponding to the two alternative folds[21–26]. Our dual-basin model for $G_A/G_B$ fold switching permits us to mimic the progression of mutations along a pathway from one fold to the other by tuning the relative interaction strengths of residue-residue contacts in the $G_A$ and $G_B$ folds (see Fig. 1b). To understand the effect of crowding, we focus on the point along the mutational pathway where the $G_A$ and $G_B$ folds exhibit roughly equal fold propensities, which we reasoned should be especially susceptible to crowding effects.

## Results

### Mimicking the mutational pathway between the $G_A$ and $G_B$ folds.

We first simulate the $G_A/G_B$ system in the absence of crowders at a fixed temperature, $T_0$, sufficiently low for low-energy folded conformations to dominate over those in the unfolded state (U). By varying the strength $\kappa_B$ of $G_B$ contacts, keeping the strength of $G_A$ contacts fixed ($\kappa_A = 1$), we can control the relative population of the two folds in our model, as shown in

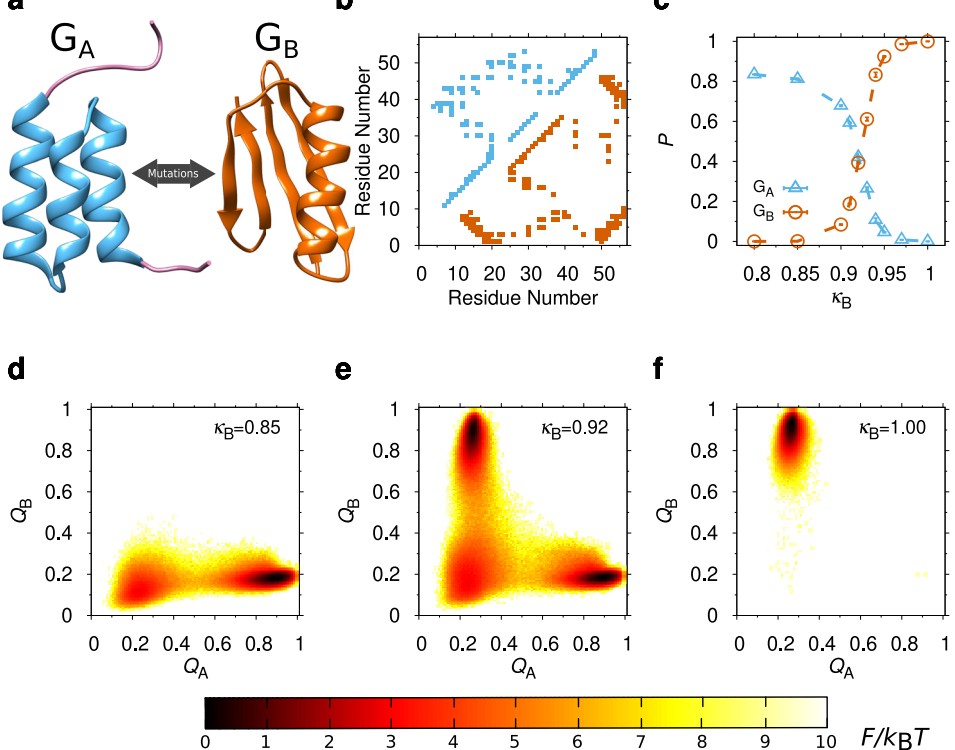

**Fig. 1 Simulating the $G_A/G_B$ fold-switch system. a** Representative experimental structures of the $G_A$ and $G_B$ folds shown in ribbon: $G_A95$ (PDB id 2KDL; blue) and $G_B95$ (PDB id 2KDM; orange). In $G_A95$, residue positions 1-7 and 53-56 are intrinsically disordered (purple). **b** Contact maps of the $G_A95$ (above diagonal) and $G_B95$ (below diagonal) structures. **c** Population $P$ of the $G_A$ (triangles) and $G_B$ (circles) folds as functions of the $G_B$ contacts strength, $\kappa_B$. **d–f** Free energy surface $F(Q_A, Q_B) = -k_B T \ln P(Q_A, Q_B)$, where $Q_A$ and $Q_B$ are the fractions of $G_A$ and $G_B$ contacts, respectively, $T$ is the temperature, $k_B$ is the Boltzmann constant, and $P(Q_A, Q_B)$ is a probability distribution, obtained at three different values of $\kappa_B$. Results in (**c**)–(**f**) are taken at temperature $T_0$ (in model units, $k_B T_0 = 0.88$, where $k_B$ is the Boltzmann constant). Error bars in (**c**) and other figures unless otherwise stated, represent $1\sigma$ standard error of the mean estimated from independent simulations.

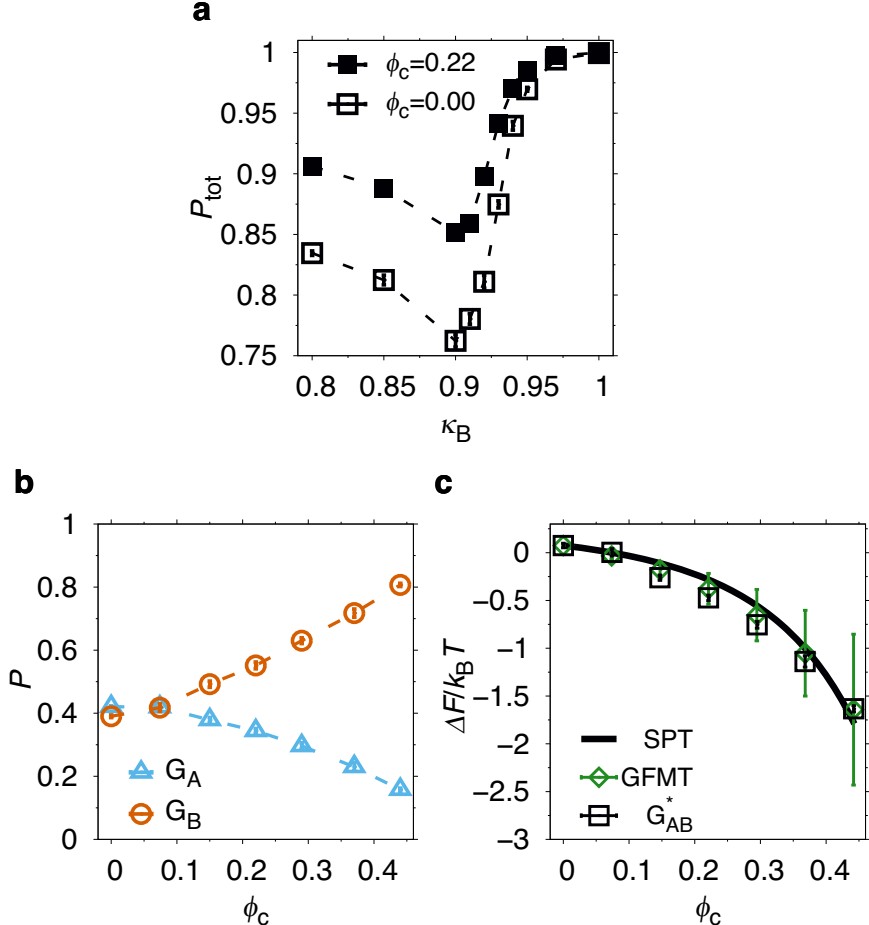

**Fig. 2 Crowding effects on total and relative fold populations. a** The total native population $P_{tot} = P_A + P_B$ as function of the contact strength $\kappa_B$ in the absence (open squares) and presence (filled squares) of crowders at packing fraction $\phi_c = 0.22$. **b** $G_A$ ($P_A$; triangles) and $G_B$ ($P_B$; circles) fold populations as functions of $\phi_c$ for model protein $G_{AB}^*$. **c** Free energy of fold switching $\Delta F_{switch} = -k_B T \ln P_B / P_A$ (squares) as function of $\phi_c$, fitted to Eq. (4) with $\delta$ as a single free parameter (solid curve). Green rhombuses are average $\Delta F_{switch}$ values calculated using GFMT (Eq. (5)) for a representative set of $G_A$ and $G_B$ conformations taken from simulations, and error bars indicate standard deviations over this set. The temperature is the same as in Fig. 1. Dashed lines between points are drawn to guide the eye.

Fig. 1c. While $G_B$ is the dominant state at high $\kappa_B$ ($\gtrsim 0.97$) $G_A$ dominates at low $\kappa_B$ ($\lesssim 0.85$), where there is also a non-zero population of U. At an intermediate value, $\kappa_B = \kappa^* = 0.92$, the populations of $G_A$ ($P_A$) and $G_B$ ($P_B$) are almost equal, $P_A \approx P_B \approx 0.39$–$0.42$. The drastic population shifts between states $G_A$, $G_B$, and U, can be seen from the free energy surfaces $F(Q_A, Q_B)$, where $Q_A$ and $Q_B$ are the fractions of formed $G_A$ and $G_B$ contacts, respectively, taken at different $\kappa_B$ values (see Fig. 1d–f).

The sharp structural transition around $\kappa_B \approx \kappa^*$ is reminiscent of experiments showing that very few mutational steps (or even a single step) is sufficient to tip the balance from $G_A$ to $G_B$, or vice versa, for carefully selected mutational pathways[15]. Moreover, the minimum in the total folded population $P_{tot} = P_A + P_B$ at $\kappa_B \approx \kappa^*$ (see Fig. 2a) is in line with the partial loss of stability seen for $G_A$ and $G_B$ sequences close to the transition point, e.g., $G_A 98$ and $G_B 98$[15], as well as for other fold switching proteins[1,27]. These results allow us to interpret $\kappa_B$ as a continuous parameter mimicking the number of steps taken along a mutational pathway connecting the $G_A$ and $G_B$ folds. The point $\kappa_B = \kappa^*$ thus represents a sequence located on the border between $G_A$ and $G_B$. Although a sequence with a perfect $G_A$ and $G_B$ population balance was not reported, it has been found for other fold switching systems, e.g., the E48S variant of RfaH[7] and the N11L

mutant of the Switch Arc protein[28]. At $\kappa_B = \kappa^*$, $P_{tot} \approx 0.82$ meaning there is a minor population of U under these conditions. It is possible to achieve a higher $P_{tot}$ while maintaining the $G_A$ and $G_B$ population balance by lowering the temperature below $T_0$ and adjusting $\kappa^*$ (see Supplementary Fig. S1). In the following, we focus our analysis on $T_0$ and refer to our $\kappa_B = \kappa^*$ model protein as $G_{AB}^*$.

**Macromolecular crowding effects on the $G_A/G_B$ fold switch.** Next we introduce spherical crowder particles with an effective radius $R_c = 12.5$ Å (see "Methods") into our simulation box, thereby probing the effect of volume exclusion on the $G_A/G_B$ switch from objects of roughly the size of the protein chain in either folded state. Because of steric repulsions, the protein chain must at all times avoid the space occupied by the crowders. Such loss of free volume typically stabilizes the native state (N) of single-fold proteins because any extended conformation in U becomes entropically disfavored relative to compact, folded conformations[29]. The same argument can be applied to each fold of a metamorphic protein. Hence, the overall stability of all folded states should increase. Indeed, as shown in Fig. 2a, the addition of crowders increases the total population $P_{tot} = P_A + P_B$ across all values of $\kappa_B$. Interestingly, poor stability is a common feature of

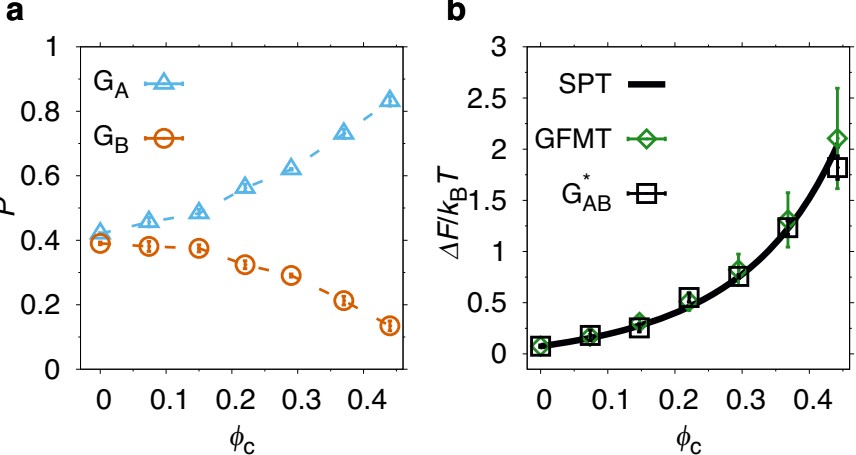

**Fig. 3 Disordered tail segments control crowding effects on the $G_A$/$G_B$ fold switch.** Results are shown for simulations with a modified potential energy function that ignores steric repulsions between any crowder and beads in chain segments 1-7 and 53-56 (see text). **a** $G_A$ (triangles) and $G_B$ (circles) fold populations $P$ as functions of $\phi_c$. **b** Fit of $\Delta F_{switch}$ (squares) to scaled particle theory (solid curve). Green rhombuses are $\Delta F_{switch}$ values calculated as in Fig. 2 but with GFMT applied only to the chain segment 8-52. The temperature is the same as in Fig. 1.

fold-switching proteins[1]. For example, sequences on either side of the $G_A$/$G_B$ switch point exhibit reduced stabilities relative to wild-type $G_A$ or $G_B$[15]. Crowding effects, if indeed providing an overall stabilization, might therefore alleviate the partial loss of stability suffered by bridge sequences in evolutionary fold-switch transitions[30].

To investigate how the relative population of the $G_A$ and $G_B$ folds is affected by crowders we focus on $G_{AB}^*$. Figure 2b shows that, as $\phi_c$ increases, the population balance exhibited by $G_{AB}^*$ at $\phi_c = 0$ swings toward $G_B$ at the expense of $G_A$, i.e., $P_B$ increases while $P_A$ decreases. The effect on $G_{AB}^*$ is not small. For example, $P_A/P_B \approx 4$ at $\phi_c = 0.44$ as compared to $\approx 1$ at $\phi_c = 0$. Hence, the effect of steric repulsions between crowders and protein is to favor to $G_B$ over $G_A$.

To quantitatively analyze this population shift we apply scaled particle theory (SPT)[31]. In this theory, the free energy cost of inserting a hard sphere of radius $R$ into a fluid of hard spheres of radii $R_c$ with packing fraction $\phi_c$ can be analytically expressed (see "Methods"). SPT has been used to model crowding-induced changes to the unfolding free energy $\Delta F_{unf} = F_U - F_N$, where $F_U$ and $F_N$ are the free energies of the U and N states, respectively[29,32,33]. Here we adapt SPT to fold switching by treating the $G_A$ and $G_B$ folds as spheres of radii $R_A$ and $R_B$. With the parametrization $R_A = R_0 - \delta$ and $R_B = R_0 + \delta$, where $R_0$ and $\delta$ are two parameters, the free energy difference can be written:

$$\beta\Delta F_{SPT} = 6\left[\left(a + 2ab + ab^2 + \frac{a^3}{3}\right)\psi + (3ab + 3ab^2 + a^3)\psi^2 + (3ab^2 + a^3)\psi^3\right], \tag{1}$$

where $a = \delta/R_c$, $b = R_0/R_c$, $\psi = \phi_c/(1 - \phi_c)$ and $\beta = 1/k_B T$. We fit the measured crowding-induced changes in free energy of fold switching, $\Delta F_{switch} = F_B - F_A = -k_B T \ln[P_B/P_A]$ to Eq. (1) with $\delta$ as a single free parameter, fixing $R_0 = R_g^{av} + \sigma_b$, where $R_g^{av} = 10.9$ Å is the average radius of gyration of the $G_A$95 and $G_B$95 native structures (see Fig. 1a), and $\sigma_b = 4.0$ Å is the radius of the beads in our protein chain. The fit is shown in Fig. 2c and gives $\delta = -0.35 \pm 0.03$ Å. The size difference $2\delta$ is in rough agreement with that calculated for the radii of gyration of the $G_A$95 and $G_B$95 native structures, $R_g^A = 11.4$ Å and $R_g^B = 10.5$ Å. The quality of the fit ($\chi^2/(n-1) = 22$, sample size $n = 7$) indicates, however, that SPT does not fully describe the observed crowding effects on $\Delta F_{switch}$. A better agreement can be obtained by

applying the generalized fundamental measure theory (GFMT) of Qin and Zhou[34,35], which takes into account both the shape of protein conformations and their fluctuations (see Fig. 2c). The generally good agreement obtained for GFMT indicates, in particular, that accounting for fluctuations in chain size, which are absent in SPT, is necessary to describe the observed $\phi_c$-dependence of $\Delta F_{switch}$.

**Disordered tails control the crowding effect on the fold switch.** The two terminal segments of the $G_A$95 structure, residues 1-7 and 53-56, are intrinsically disordered (see Fig. 1a). Hence, the $G_A$-to-$G_B$ fold switch involves a disorder-order transition of these tail regions. Given their flexible nature, it is likely that the tails contribute substantially to the volume excluded by the protein when occupying the $G_A$ fold. Indeed, if the terminal segments are ignored, the radius of gyration of $G_A$95 is reduced by $\approx 22\%$, $R_g^{A,8-52} = 8.9$ Å. By contrast, the radius of gyration of $G_B$95 determined over the same segment is $R_g^{B,8-52} = 10.8$ Å, which is a slight increase compared to value for the full chain. Together with the poor fit with SPT (see Fig. 2c), these results suggest a potential role for the tail segments in how the $G_A$/$G_B$ fold switch is impacted by crowding.

To show that this is indeed the case, we carry out crowding simulations with a modified potential energy function, $E_{mod}^{(db)}$, in which all crowder-protein interactions have been turned off for residues in the 1-7 and 53-56 regions. Hence, in these simulations, the N- and C-terminal segments become invisible to the crowders, which thus freely overlap with the residues. Although unphysical, this computational experiment logically tests the role of the tail regions in our model under crowded conditions. Note that crowders can overlap with the tails regardless of which state is populated by the protein. Moreover, at $\phi_c = 0$, the model remains the same because intra-chain interactions are unaffected. The results are shown in Fig. 3a. Strikingly, with the modified potential $E_{mod}^{(db)}$, the impact of crowding reverses such that the $G_A$ fold becomes increasingly favored over $G_B$ with increasing $\phi_c$. Moreover, the fit to SPT (see Fig. 3b), obtained using $R_0 = 13.9$ Å and giving $\delta = 0.42 \pm 0.01$ Å, is now much better ($\chi^2/(n-1) \approx 1.5$). We also carried out simulations with the modifications in $E_{mod}^{(db)}$ applied separately to the 1-7 and 53-56 regions. As it turns out, the effects on $\Delta F_{switch}$

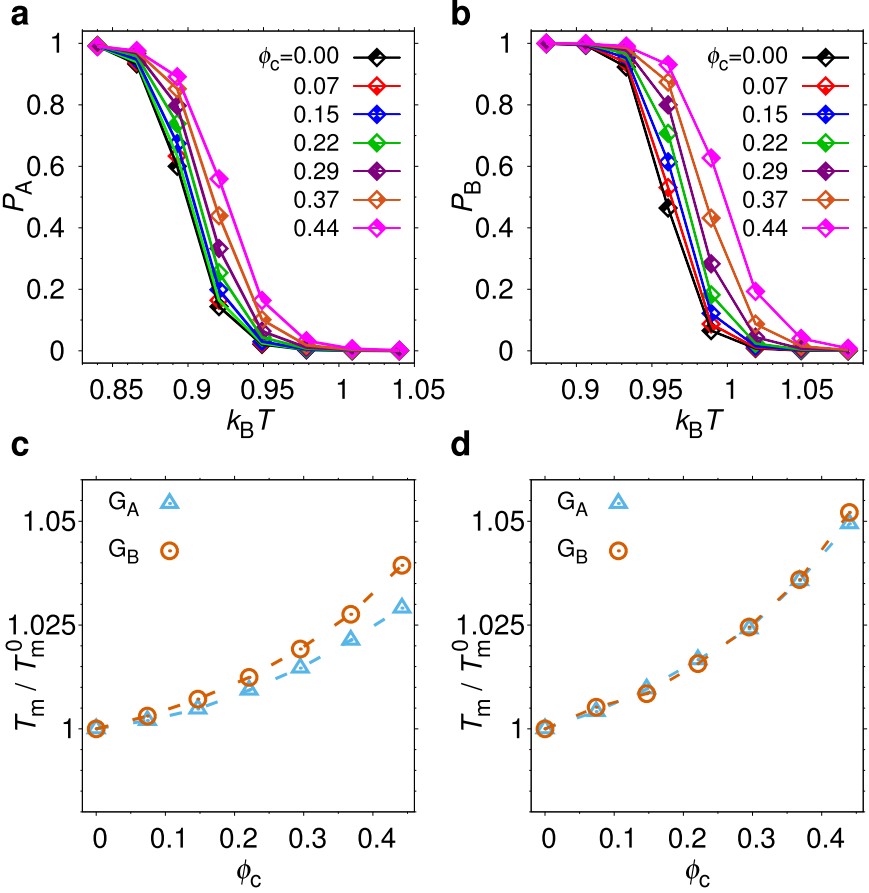

**Fig. 4 Crowding effects on single-fold $G_A$ and $G_B$ proteins. a** $G_A$ fold population, $P_A$, obtained with our dual-basin structure-based model with weak $G_B$ contacts ($\kappa_B = 0.85$), as function of temperature. **b** $G_B$ fold population, $P_B$, obtained with the same model but with strong $G_B$ contacts ($\kappa_B = 1.00$). **c** Midpoint temperature, $T_m$, as function of $\phi_c$. $T_m$ is obtained by fitting the folding curves in (**a**) and (**b**) to a two-state model. **d** $T_m$ as function of $\phi_c$, obtained with single-basin structure-based models for $G_A$ and $G_B$. In both (**c**) and (**d**), $T_m^0$ is the value of $T_m$ at $\phi_c = 0$.

is roughly additive (see Supplementary Fig. S2), suggesting that the two tail regions independently reduce the volume available to the crowders. Taken together, our computational experiment shows that the volume excluded by the disordered tails in the $G_A$ fold is the dominant factor affecting the balance between the folds in the presence of crowders.

**Comparing with crowding effects on single-fold proteins.** Above we have shown that the crowders induce a population shift in $G_{AB}^*$, which is due to the presence of disordered tails. For single-fold (monomorphic) proteins, purely repulsive crowders typically enhance the stability of N[36]. Naively, one may therefore expect that N of monomorphic $G_B$ ($\kappa_B > \kappa^*$) would be more strongly stabilized by the crowders than monomorphic $G_A$ ($\kappa_B < \kappa^*$). To test this idea, we determine the folding midpoint temperature, $T_m$, for the model proteins with $\kappa_B = 0.85$, which adopts the single fold $G_A$, and $\kappa_B = 1.00$, which adopts the single fold $G_B$ (see Fig. 1), over a range of $\phi_c$. As seen in Fig. 4a–c, both proteins exhibit a monotonic increase in $T_m$ with increasing $\phi_c$, indicating stabilization. The relative increase in $T_m$ for monomorphic $G_B$ is indeed somewhat larger than for monomorphic $G_A$. The difference is relatively small, however. We also perform similar simulations using the single-basin energy functions $E^{(A)}$ and $E^{(B)}$, (see "Methods") which lack entirely a bias toward the alternative fold. For these models, the crowding-induced increases in $T_m$ are almost identical (see Fig. 4d). Taken together, these results suggest that determining the crowding response of a fold switcher with two "co-existing" folds may not be easily obtained

from experiments on single-fold proteins representative of the two folds.

**The unfolded state changes character across the fold switch.** The results in Fig. 4 are at first surprising because $\Delta F_{switch}$ for a fold switching protein can be obtained from the relation:

$$\Delta F_{switch} = \Delta F_{unf}^A - \Delta F_{unf}^B, \qquad (2)$$

where $\Delta F_{unf}^A = F_U - F_A$ and $\Delta F_{unf}^B = F_U - F_B$ are defined in direct analogy with the unfolding free energy of a single fold protein. Equation (2) expresses that a decrease in $\Delta F_{switch}$ results when the crowding-induced stabilization of fold $G_B$ relative to U is stronger than the stabilization of fold $G_A$. However, Eq. (2) is only guaranteed to hold when $\Delta F_{switch}$, $\Delta F_{unf}^A$ and $\Delta F_{unf}^B$ are determined for the same protein for which U provides a common reference. We therefore examine if the drastic structural shift for low-energy (folded) conformations in the $G_A/G_B$ fold switch is accompanied by changes in U.

We first characterize U across the fold switch in the absence of crowders, i.e., upon changing the contact strength $\kappa_B$, as shown in Fig. 5a, b. With increasing $\kappa_B$, and therefore increasing $G_B$ population, the unfolded state radius of gyration $R_g^{(U)}$ decreases. Additionally, U becomes more "$G_B$-like" as shown by the increase in $Q_B^{(U)}$, i.e., the fraction of formed $G_B$ contacts in U. These results are in line with simulations of single-fold proteins showing that native contacts in $\beta$-proteins tend to promote chain collapse during folding more efficiently than $\alpha$-proteins[37]. In the

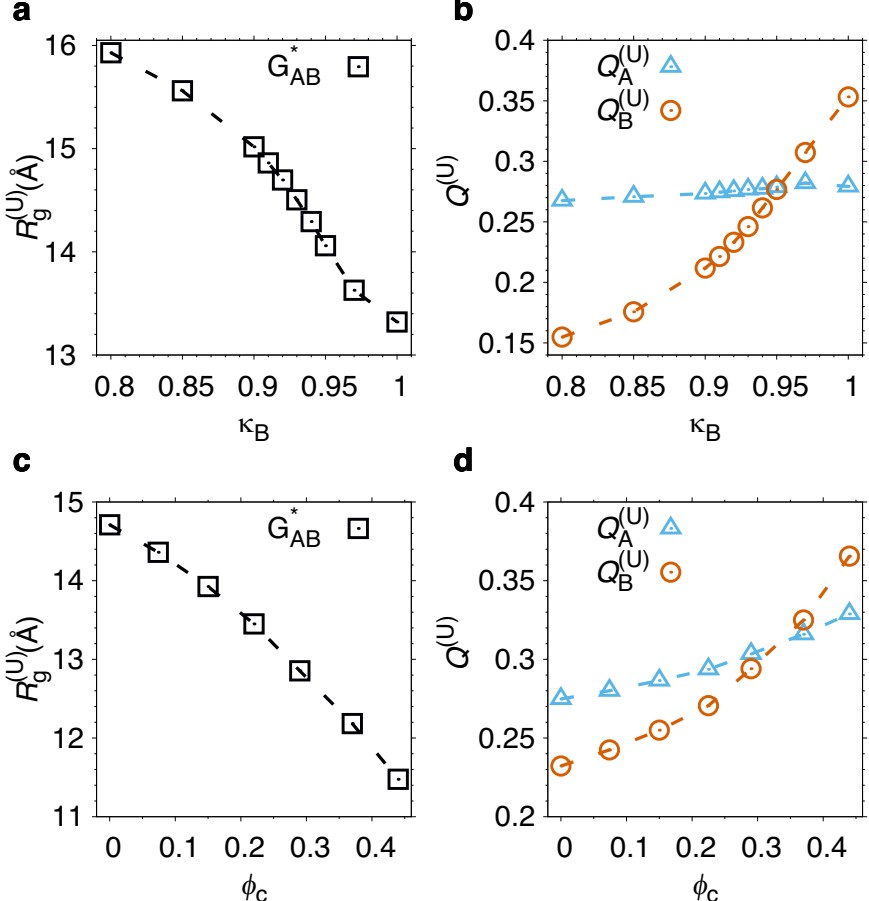

**Fig. 5 Changes to the unfolded state character across the fold switch.** a $R_g^{(U)}$, b $Q_A^{(U)}$ (triangles) and $Q_B^{(U)}$ (circles) as functions of the contact strength $\kappa_B$ at $\phi_c = 0$, where $R_g^{(U)}$, $Q_A^{(U)}$, and $Q_B^{(U)}$ are the radius of gyration, fraction of $G_A$ contacts, and fraction of $G_B$ contacts, respectively, determined for the unfolded state, U. c $R_g^{(U)}$, d $Q_A^{(U)}$ (triangles) and $Q_B^{(U)}$ (circles) as functions of $\phi_c$, obtained for $G_{AB}^*$ ($\kappa_B = 0.92$). The temperature is the same as in Fig. 1.

crowding-induced $G_A$-to-$G_B$ fold switch we similarly find a compaction of U (see Fig. 5c, d). For $\phi_c > 0.20$, $R_g^{(U)}$ becomes smaller than for any value of $\kappa_B$ in the case of no crowders. Moreover, $Q_A^{(U)}$ and $Q_B^{(U)}$ both increase with $\phi_c$. In summary, fold switching driven either by mutation or crowding substantially impacts the structural characteristics of U. Both chain compaction and the formation of residual structure due to crowding have been observed for various single-fold proteins[38,39].

## Discussion
Fold switching in proteins involves major structural changes, including in shape and amino acid composition of surface regions. As a result, fold switching should be inherently susceptible to crowding effects. Here we tested this idea by applying a dual-basin structure-based protein model and purely repulsive crowders to the $G_A/G_B$ fold switch. We found that the addition of crowders indeed alters the free energy balance between the two folds. The effect increases monotonically with $\phi_c$. At $\phi_c = 0.44$, the change in $\Delta F_{switch}$ is $\approx 2k_B T$ in magnitude. While no experiment probing crowding effects on the $G_A/G_B$ fold switch is available for comparison, a key role for molecular shape in crowding has been demonstrated in a study that exploited alternative dimer forms of two almost identical sequences[40]. Very recently, it was shown using nuclear magnetic resonance spectroscopy that the addition of 90 g/l Ficoll, polyethylene glycol or BSA to the solution impacted the relative fold population of the two metamorphic proteins KaiB and XCL1[41].

Our results show that crowding effects on the $G_A/G_B$ system may be determined by chain segments at the N- and C-terminal ends, which are intrinsically disordered only in the $G_A$ fold. The volume excluded by these disordered segments leads to an entropic stabilization of $G_B$ relative to $G_A$. Interestingly, order-disorder transitions occur frequently in protein fold switching[1]. One example besides $G_A/G_B$ is human chemokine XCL1, which switches folds upon dimerization. In its monomeric (chemokine) fold, XCL1 adopts an $\alpha$-helix in its C-terminal region, which becomes disordered when the protein transforms to its dimeric fold-switched state[12]. It should be pointed out that crowder interactions other than hard-core steric repulsions can modify the crowding effects. For single-fold proteins, nonspecific attractive (soft) interactions between protein and crowders generally counteract the stabilizing effect of volume exclusion[42], and can even lead to a net destabilization[43].

Most studies on fold switching have quite naturally focused on the structure and dynamics of the different folded states, and their interconversions. However, our simulations of the $G_A/G_B$ switch reveal that fold switching may be accompanied by substantial changes in U (see Fig. 5). Under conditions favoring $G_A$, we find that U is rather expanded and dominated by local contacts while becoming more compact and forming more non-local contacts as the conditions shift to favor $G_B$. In previous simulations of the metamorphic protein RfaH[25], we showed that the isolated C-terminal domain, which adopts a stable $\beta$-barrel in isolation, exhibits a propensity for $\alpha$-helical structure in U. This helical propensity was demonstrated experimentally by Zuber et al.[27],

who suggested further that the presence of residual helical structure may help initiate the reverse fold switching of RfaH, i.e., the transformation from the $\beta$-barrel to its alternative all-$\alpha$ fold. Taken together, the above considerations suggest that an improved understanding of the unfolded state of metamorphic proteins may give further insights into fold switching mechanisms, as well as effects from crowding.

In addition to changes to the relative population of the two folds, we have found that the presence of crowders increases the total population of the $G_A$ and $G_B$ folds relative to U. An overall stabilization of ordered states might be especially beneficial to fold-switching proteins, which often exhibit reduced stabilities[1]. Poor stabilities of bridge sequences at the border between folds may hamper evolutionary transitions[16,44,45]. A recent study suggests fold switching within the context of multidomain proteins, in which non-switching domains can act as stabilizing scaffolds, may help stabilize such bridge sequences and facilitate fold transitions[13]. Our results suggest that additional stabilization may be provided by crowding effects.

Our study opens up for additional experimental and theoretical investigations into the effects of crowding on fold switching. Recent advances in the fold switching field are improving our understanding of this phenomenon within functional[27,46] and evolutionary[3,12,13] contexts. These efforts should also include a characterization of the impact of crowding effects on equilibrium and kinetic properties of fold switching proteins.

## Model and methods

**Native structures and contact maps.** The experimentally determined structures of $G_A 95$ (PDB id 2KDL) and $G_B 95$ (2KDM)[15] were downloaded from the Protein Data Bank. Contact maps for 2KDL and 2KDM were obtained as prescribed by the shadow map method[47] and contained 106 and 145 contacts, respectively.

**Observables.** The fractions of native contacts were determined using $Q_A = N_A/106$ and $Q_B = N_B/145$, where $N_A$ ($N_B$) is the number of $G_A$ ($G_B$) contacts formed. A contact between two amino acids i and j was considered formed if $r_{ij} < 1.2 r_{ij}^0$, where $r_{ij}$ is the distance between the $C_\alpha$ atoms and $r_{ij}^0$ is the distance in the native structure (2KDL or 2KDM). In determining the fold populations, $P_A$ and $P_B$, we classified a conformation to be in the $G_A$ fold if $N_A > N_A^{cut} = 58$ and in the $G_B$ fold if $N_B > N_B^{cut} = 76$, where $N_A^{cut}$ and $N_B^{cut}$ were determined based on the free energy profiles $F(N_A)$ and $F(N_B)$ for $G_{AB}^*$ (see Supplementary Fig. S3).

**Coarse-grained model for protein fold switching.** Simulations were carried out using a dual-basin structure-based model in which each amino acid is represented by a single bead located on the $C_\alpha$-atom position. The starting point for developing this model was a modified version of the single-basin structure-based model developed previously[18]. The single-basin model has a potential energy function with 5 terms, $E = E_{bond} + E_{bend} + E_{torsion} + E_{rep} + E_{cont}$, representing bond stretching, bond flexing, torsional rotations, repulsions between bead pairs, and attractive native contact interactions. We applied this model separately to the native structures of $G_A 95$ and $G_B 95$ resulting in two structure-based energy functions, $E^{(A)}$ and $E^{(B)}$, with single basins of attraction (either the $G_A$ fold or the $G_B$ folds). Using the exponentially-weighted mixing approach of Best et al.[48], we then merged $E^{(A)}$ and $E^{(B)}$ into a single (dual basin) energy function, $E^{(db)}$. The strength of $G_A$ and $G_B$ contacts, $\kappa_A$ and $\kappa_B$, were left as free parameters in $E^{(db)}$, allowing the relative depth of the $G_A$ and $G_B$ basins of attraction to be controlled. Full details of the model are given in the Supporting Information (see Supplementary Notes).

**Excluded volume crowders.** Crowder-crowder and crowder-protein interactions are described using the potential function[29]:

$$V(r) = \epsilon \left( \frac{\sigma}{r - \rho + \sigma} \right)^{12} \qquad (3)$$

for distances $r > \rho - \sigma$, and $V(r) = \infty$ otherwise. Hence, our crowders have a soft repulsive shell over a hard core. The parameters $\rho$ and $\sigma$ control the range of the interaction and the width of the soft repulsive shell, respectively. We determined these parameters using $\sigma = \sigma_i + \sigma_j$ and $\rho = R_i + R_j$, where i and j are two interacting elements. When i, j are crowders we set $\sigma_i = \sigma_j = 3$ Å and $R_i = R_j = 12$ Å, and when one of i, j is a crowder and the other is a chain bead we set for the bead (assuming j) $\sigma_j = R_j = \sigma_b$, where $\sigma_b = 4$ Å is the bead radius. With this choice of $\rho$ and $\sigma$, an approximate value for the crowder radius $R_c$ is $\approx 12$ Å. A more precise value was obtained from the radial distribution function $g(r)$ for a large crowder-only system, which indicated 12.5 Å (see Supplementary Fig. S4). We therefore use $R_c = 12.5$ Å for the crowder radius throughout this work. The crowder concentration, as quantified by the fraction of the total simulation volume $V$ occupied by the crowders, is then given by $\phi_c = 4\pi R_c^3 N_c / 3V$. In our simulations, the number of crowder particles $N_c$ ranges from 9 for $\phi_c = 0.073$ to 54 for $\phi_c = 0.442$.

**Langevin dynamics.** Conformational sampling was carried out using Langevin dynamics following our previous approach[18]. The time evolution of the system is then governed by the equation, $m\dot{v}(t) = F_{conf} - m\gamma v(t) + \eta(t)$, where $m$, $v$, $\dot{v}$, $\gamma$, $F_{conf}$ and $\eta(t)$ are the mass, velocity, acceleration, friction coefficient, conformational force and random force, respectively. The random force $\eta(t)$ is drawn from a Gaussian distribution, the variance of which sets the temperature of the system. For computational reasons, simulations were carried out in the low-friction (underdamped) limit, where $-m\gamma v(t)$ is small relative to the inertial term $m\dot{v}(t)$. In this limit, a natural unit of time for the dynamics is $\tau = \sqrt{ml^2/\epsilon}$[49], where $\epsilon$ is the magnitude of typical interactions and $l$ is a length scale, which we set to 4 Å. The friction coefficient for beads was taken to be $\gamma_b = 0.05\tau^{-1}$. Units were set so that the mass of a bead is $m_b = 1.0$. Numerical integration of the equation of motion was carried out using the velocity form of the Verlet algorithm[50] with an integration time step $\delta t = 0.005\tau$. For crowders, the mass and friction coefficient were set to $m_c = 9.0$ and $\gamma_c = 0.017\tau^{-1}$.

**Simulation and analysis details.** Simulations were carried out by placing the protein and crowders in a cubic box with side 100 Å. Periodic boundary conditions were applied. Langevin dynamics simulations were used to determine the equilibrium behavior of various systems characterized by different $G_B$ contact strengths $\kappa_B$ and crowder concentrations $\phi_c$. Simulations were performed at either fixed temperature or using simulated tempering[51], in which temperature changes dynamically between a predetermined set of values. In the simulated tempering runs, temperatures were updated every 100 time steps. For each system, 5-10 independent runs of $(4 - 5) \times 10^9$ time steps each were carried out and used to estimate averages and statistical uncertainties. All simulations were initiated from a random protein conformation (random torsional angles $\phi_i$) and random crowder positions, followed by a Monte Carlo-based relaxation step in which all hard core steric clashes were removed.

**Theory.** Simulation results were analyzed using scaled particle theory[31] and generalized fundamental measure theory (GFMT)[34,35]. According to SPT, the free energy cost of inserting

a hard sphere of radius $R$ in a hard sphere fluid of particles with radius $R_c$ is[31]:

$$\beta F = (3x + 3x^2 + x^3)\psi + \left(\frac{9x^2}{2} + 3x^3\right)\psi^2 + 3x^3\psi^3 - \ln(1 - \phi_c), \quad (4)$$

where $\beta = 1/k_B T$, $T$ is the temperature, $k_B$ is the Boltzmann constant, $x = \frac{R}{R_c}$, $\psi = \frac{\phi_c}{1-\phi_c}$, and $\phi_c$ is fluid volume fraction. Minton showed that SPT predicts a strong stabilizing effect on the native states of single-fold proteins if U is modeled as an ideal Gaussian chain[33]. Here SPT was used to model the $G_A/G_B$ folds as spheres of different radii.

GFMT accounts for geometric features of the protein structure using three quantities, linear size $l_p$, surface area $s_p$, and volume $v_p$. These quantities are obtained by sampling the crowder-excluded surface, which is influenced by the crowder radius, chain bead radius, and the protein conformation. The free energy cost of inserting a protein conformation into a crowder fluid is then estimated using[34,35]:

$$\beta F = \beta \Pi_c \nu_p + \beta \gamma_c s_p + \beta \kappa_c l_p - \ln(1 - \phi_c), \quad (5)$$

where $\Pi_c$, $\gamma_c$ and $\kappa_c$ are osmotic pressure, surface tension, and bending rigidity (curvature), respectively, of the crowder fluid. To calculate the change in fold switching free energy at different $\phi_c$ using GFMT, we set the radii of crowder and beads to 12.5 and 4.0 Å, respectively. GFMT calculations were performed on 200 random chain conformations for each of the $G_A$ and $G_B$ folds, taken from our simulations of the model protein $G_{AB}^*$ at temperature $T_0$.

## Data availability

Data sharing not applicable to this article as no datasets were generated or analysed during the current study.

## Code availability

Simulations were carried out using in-house software written in C. The software will be made available for academic use upon request to the corresponding author.

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

## Acknowledgements

S.W. acknowledges support from the Natural Sciences and Engineering Research Council of Canada (grant RGPIN-2016-05104). This research was enabled in part by the computational resources provided by the Digital Research Alliance of Canada.

## Author contributions

S.W. designed the study. S.B., B.S. and S.W. developed the model, carried out the simulations, performed the analyses, and wrote the paper.

## Competing interests

The authors declare no competing interests.
