## [Peer Review File · Communications Chemistry]

Reviewers' comments:

Reviewer #1 (Remarks to the Author):

The study investigated the effects of macromolecular crowding on fold switching, with a dual-basin model based on GA95 (PDB id 2KDL) and GB95 (PDB id 2KDM). The proteins are represented with 56 beads at Calpha. The radii for beads and spheric crowders are 4 and 12 Å, with soft repulsiveness. The simulation and result look reasonable. However, the application of scaled particle theory (SPT) is very questionable. The equation looks correct, but the fitting results are incorrectly interpreted. For figure 2c, the fitting required a difference of 1.5 Å between A and B, while the real difference is only 0.9 Å, given $R_A g = 11.4$ Å and $R_B g = 10.5$ Å, as reported. The 0.6 Å is significant, as shown in left panel in the pdf attached. directly using the R_g will bring the black line (fitted) to the red line (R_g). On the other hand, using R_g is a very poor approximation for folded protein, as the R_g are calculated with the center of beads, ignoring the radius of the bead. Using R_g+4 (blue line) could fix the problem apparently but will cause problems for fig3b. The authors didn't state the fitting parameter for fig3b, but I would guess it's not close to the change on R_g . I used R_g from the first model in the pdb to generate the red and blue line.

For coarse-grained representation proteins with crowders, the Generalized fundamental measure theory (GFMT), are more precisely, GFMT accounts the linear size, surface area, and volume of protein, comparing radius only in SPH. GFMT even works with unfolded states, as shown in

<https://iopscience.iop.org/article/10.1088/1478-3975/10/4/045001>

<http://web2.physics.fsu.edu/~zhou/reprints/pb187.pdf>

As GFMT calculates the value based on protein conformation, it's much less arbitrary than SPH. Attached right panel in the pdf attached shows the result from GFMT with 20 models in the pdb. The high half matches well to fig3b; the low half has a much larger error bar, showing the flexibility of the tail. Given the simulated structure could be more diverse than that in pdb, the mean could shift.

The GFMT are described in <https://journals.aps.org/pre/abstract/10.1103/PhysRevE.81.031919>

<http://web2.physics.fsu.edu/~zhou/reprints/pr142.pdf>

The code are also viable at <http://pipe.rcc.fsu.edu/gfmt/gfmt.tar.gz>

In total, the good fitting in SPT without tail only indicated that the data are less fluctuating. So, the authors should at least correctly state the problem with SPT, or reanalysis the result with GFMT. With both 2 folded states and an unfold state, the result could be more understandable.

Minor points,

1. for residues turned off in the 1-7 and 53-56 regions, but report $R_A, 7-50 g = 8.2$ Å, so 8-52 and 7-50 is slightly inconsistent. Also, Both R_A and R_B without tail should be reported, as well as the fitting parameter from SPH to show the problem of SPH.
2. The P for GA and GB change slightly at 0, is there any reason?

Reviewer #2 (Remarks to the Author):

In the study, the researchers used Coarse-Grained Molecular Dynamics (CG-MD) simulations to investigate how macromolecular crowding affects the behavior of fold-switching proteins. They also compared their findings to the predictions made by Scaled Particle Theory (SPT). The results of the research were deemed to be of high quality and suitable for publication.

However, there were a few concerns that the authors should address before submitting their work for publication. First, they needed to provide a justification for their choice of $R_c = 12 \text{ \AA}$. This refers to the radius of the crowding particles in the simulation. It is important to explain why this particular value was chosen and how it relates to the goals of the study.

Second, the authors did not provide a clear explanation of how the population value (P) was calculated. It is necessary to describe the method used to calculate P and provide enough detail to enable other researchers to replicate the results.

Finally, the temperatures corresponding to figures 1, 2, 3, and 5 were not specified. The authors should provide this information in the manuscript or in the figure captions.

Reviewer #3 (Remarks to the Author):

The article by Bazmi, Seifi and Wallin, presents a computational exploration of the effects of macromolecular crowding on the fold-switch of the GA and GB folds, a system that has been engineered to reach very high sequence identity while still maintaining structurally different native states. The foundations of the manuscript are quite interesting, which is to explore macromolecular crowding effects that are closer to cellular condition, given that most experiments have been carried out in diluted solutions.

The results are compelling, suggesting that each fold in a fold-switching protein responds differently to crowding effects, such that it can produce shifts in the balance between folds and dictating a preferential stability for one over the other. The authors also delivered control simulations using the single-basin models to confirm the effects of macromolecular crowders.

These results might call the intention of experimentalist to work on the effects of macromolecular crowders on fold-switching, although the GA and GB system might not be the most appropriate one for such endeavor, given that, as the authors mention on page 5, a sequence with a perfect GA and GB population balance has not been reported. Yet, the authors do mention other metamorphic proteins that might be suitable for such studies.

Major Comments:

1) The only major comment that I have is regarding the model in which the tails of GA are impeded to interact with the macromolecular crowders. I am wondering whether the overlap allowed by the removal of the crowding effects in these regions leads to stabilization of the GA fold over the GB fold due to a favorability of a fold with a smaller radius of gyration (here, GA).

Given that the disordered regions in the GA fold have different extensions, what would happen if the crowder-protein interactions are systematically deleted for the N-termini and C-termini alone? Are these effects additive?

Minor Comments:

1) Regarding the dual-basin approach, I am wondering whether the value for the strength of native contacts for the GB fold has anything to do with the proportion of native contacts in the GA and GB folds. Was this value chosen by exploration at random or by looking at the number of native contacts of each fold? Intuitively, the sum of the strength of all native contacts in a given fold should be equal to the sum of the strength of all native contacts in the alternative fold in order to achieve isoenergetic basins, but it could also be that the geometry of the native interactions due to the topology of each fold has an effect on the energetics.

2) On the same note, what drove the decision to keep the strength of GA contacts fixed instead of GB?

3) The free energy landscapes and populations for each fold are presented at a fixed temperature T in Fig. 1, similar to the total folded populations in Fig. 2. Yet, I wonder whether shifting the temperature to slightly lower or higher values is sufficient to achieve equal populations of each fold and higher values for total folded populations at different strengths of native contacts for the GB fold. Could the authors elaborate whether the plots being presented correspond to the temperature at which the populations of each fold reach its maximum? If not, this would be a reasonable explanation for the minimum P_{tot} value in Fig. 2.

4) On the same note, it might be possible that the stabilization of GB over GA in the presence of crowders could be due to shifts in the temperature at which both states are under equilibrium. Could the authors produce the plots of the populations of GA and GB as a function of temperature for each crowder packing fraction and for the simulations in the absence of crowders? This would be an indication of stabilization as the authors did for the single-fold systems in Fig. 4, where they checked the increased in T_m due to the presence of crowders.

4) What is the rationale for the choice of crowder particles of radius 12 \AA ? How is this size comparable to the radius of the beads in the coarse-grained model?

5) Which software/simulation package was used for these simulations?

We thank the reviewers for their thoughtful comments on our manuscript, which we feel have been strengthened. We would like to highlight two changes: (1) Prompted by the comments of Reviewer 1, we have determined a new value for the crowder radius (12.5 Å). As a result, crowder volume fractions, ϕ_c , have increased by a factor $(\frac{12.5}{12})^3 \approx 13\%$. This change does not impact any of our conclusions. However, numerical values of ϕ_c and all figures that include ϕ_c -dependent quantities, i.e., Figs. 2a-c, 3a-b, 4a-d, and 5c-d, have been updated; (2) We have inserted a comment in our Discussion (on page 10) on an article (N. Zhang, W. Guan, S. Cui, and N. Ai. Crowded environments tune the fold-switching in metamorphic proteins. Commun Chem, 6:117, 2023) that appeared after submission of our manuscript. This new experimental work (new Ref. [41]) confirms that crowding can indeed impact the population balance between folds in fold-switching proteins. We feel that this experimental work, which focuses on other metamorphic proteins, strengthens our work.

Below follow our detailed responses to each of the points raised.

Reviewer 1

The study investigated the effects of macromolecular crowding on fold switching, with a dual-basin model based on GA95 (PDB id 2KDL) and GB95 (PDB id 2KDM). The proteins are represented with 56 beads at Calpha. The radii for beads and spheric crowders are 4 and 12 Å, with soft repulsiveness. The simulation and result look reasonable. However, the application of scaled particle theory (SPT) is very questionable. The equation looks correct, but the fitting results are incorrectly interpreted. For figure 2c, the fitting required a difference of 1.5 Å between A and B, while the real difference is only 0.9 Å, given $R_A = 11.4$ Å and $R_B = 10.5$ Å, as reported. The 0.6 Å is significant, as shown in left panel in the pdf attached. directly using the R_g will bring the black line (fitted) to the red line (R_g). On the other hand, using R_g is a very poor approximation for folded protein, as the R_g are calculated with the

center of beads, ignoring the radius of the bead. Using R_g+4 (blue line) could fix the problem apparently but will cause problems for fig3b. The authors didn't state the fitting parameter for fig3b, but I would guess it's not close to the change on R_g . I used R_g from the first model in the pdb to generate the red and blue line.

For coarse-grained representation proteins with crowders, the Generalized fundamental measure theory (GFMT), are more precisely, GFMT accounts the linear size, surface area, and volume of protein, comparing radius only in SPH. GFMT even works with unfolded states, as shown in

<https://iopscience.iop.org/article/10.1088/1478-3975/10/4/045001>

<http://web2.physics.fsu.edu/~zhou/reprints/pb187.pdf>

As GFMT calculates the value based on protein conformation, it's much less arbitrary than SPH. Attached right panel in the pdf attached shows the result from GFMT with 20 models in the pdb. The high half matches well to fig3b; the low half has a much larger error bar, showing the flexibility of the tail. Given the simulated structure could be more diverse than that in pdb, the mean could shift. The GFMT are described in

<https://journals.aps.org/pre/abstract/10.1103/PhysRevE.81.031919>

<http://web2.physics.fsu.edu/~zhou/reprints/pr142.pdf>

The code are also viable at <http://pipe.rcc.fsu.edu/gfmt/gfmt.tar.gz>

In total, the good fitting in SPT without tail only indicated that the data are less fluctuating. So, the authors should at least correctly state the problem with SPT, or reanalysis the result with GFMT. With both 2 folded states and an unfold state, the result could be more understandable.

Our response: We have re-analyzed our data using GFMT. We found first (as you did) that neither SPT nor GFMT provides a perfect agreement with our data in all instances. However, we realized that the precise value of the crowder radius influences both theories through the packing fraction ϕ_c as well as through the crowder-excluded surface in the case of GFMT. We therefore determined a more accurate value for R_c from the pair correlation

function $g(r)$ obtained from simulations of a crowder-only system (see the new Figure S4). The effective crowder radius turned out to be 12.5 \AA , slightly larger than the original value (12 \AA).

With this new value of R_c , we find a good agreement with GFMT for both our regular dual-basin model $E^{(db)}$ (Figure 2c) and the modified version $E_{mod}^{(db)}$ (Figure 3b). The quality of the SPT fit in Figure 2c remains rather poor. Hence, our revised analysis, which uses both SPT and GFMT, highlights the importance of both chain shape and fluctuations for describing our data.

The new GFMT results are described in section 2.2 on page 6 and in the figure legends of Figures 2c and 3b. A brief description of GFMT was added in Methods (see section 4.7 on page 15). The following references were inserted:

[34] S. Qin and H. X. Zhou. Generalized fundamental measure theory for atomistic modeling of macromolecular crowding. *Phys Rev E*, 81:031919, 2010.

[35] S. Qin, J. Mittal, and H. X. Zhou. Folding free energy surfaces of three small proteins under crowding: validation of the postprocessing method by direct simulation. *Phys Biol*, 10:045001, 2013.

Section 4.4 was modified to reflect the new meaning of R_c as an effective radius. Note that the crowder potential function $V(r)$ (Eq. (3)), and its parameters, have not changed. We have merely determined a more accurate value of the radius of our crowders.

A small typo in Eq. 1 was corrected.

Minor points,

1. for residues turned off in the 1-7 and 53-56 regions, but report $RA, 7-50 g = 8.2 \text{ \AA}$, so 8-52 and 7-50 is slightly inconsistent. Also, Both RA and RB without tail should be reported,

as well as the fitting parameter from SPH to show the problem of SPH.

Our response: Thank you. The correct segment is indeed 8-52. We have corrected this. The numbers have changed slightly but the trends remain the same. Additionally, as suggested, we now report R_g values taken both over the entire chain 1-56 and over the 8-52 region for both GA95 and GB95. Please see updated text in section 2.3 on page 7.

2. The P for GA and GB change slightly at 0, is there any reason?

Our response: There were indeed slight differences between the estimates of P_A and P_B at $\phi_c = 0$ in Figure 2b and Figure 3a, because they were obtained from two different sets of simulations. The results were fully consistent, however. To avoid confusion, we now report results from only one of the simulation sets. Figure 3 have been updated to reflect the change.

Reviewer 2

In the study, the researchers used Coarse-Grained Molecular Dynamics (CG-MD) simulations to investigate how macromolecular crowding affects the behavior of fold-switching proteins. They also compared their findings to the predictions made by Scaled Particle Theory (SPT). The results of the research were deemed to be of high quality and suitable for publication.

However, there were a few concerns that the authors should address before submitting their work for publication. First, they needed to provide a justification for their choice of $R_c = 12 \text{ \AA}$. This refers to the radius of the crowding particles in the simulation. It is important to explain why this particular value was chosen and how it relates to the goals of the study.

Our response: Our goal was to probe the role of hard-core repulsions (excluded volume) between macromolecular crowders and the protein. It is expected that the excluded volume effect becomes important when the crowders are comparable in size or smaller than the protein [e.g. Rivas and Minton, Trends Biochem Sci 41 2016]. Hence, to study macromolecular

crowding effects on G_A/G_B we picked a crowder radius that is larger than the chain beads ($R_{\text{bead}} = 4 \text{ \AA}$) but comparable to the GA and GB native structures. We have clarified our choice of crowders in the beginning of section 2.2 on page 5.

Second, the authors did not provide a clear explanation of how the population value (P) was calculated. It is necessary to describe the method used to calculate P and provide enough detail to enable other researchers to replicate the results.

Our response: The fold populations P_A/P_B were calculated based on a cutoff in the number of GA/GB contacts formed. Specifically, the GA fold is considered formed if the # GA contacts > 58 and the GB fold is considered formed when the # GB contacts > 76 . The cutoff values, 58 and 76, were determined based on the barrier locations in the free energy profiles obtained for G_{AB}^* . The choice of cutoff values are now explained in section 4.2 on page 12 and in the new Figure S3.

Finally, the temperatures corresponding to figures 1, 2, 3, and 5 were not specified. The authors should provide this information in the manuscript or in the figure captions.

Our response: All simulation results in these figures were taken at the lowest studied temperature T (in model units, $k_B T = 0.88$). We have now added this information to the respective figure captions.

Reviewer 3

The article by Bazmi, Seifi and Wallin, presents a computational exploration of the effects of macromolecular crowding on the fold-switch of the GA and GB folds, a system that has been engineered to reach very high sequence identity while still maintaining structurally different native states. The foundations of the manuscript are quite interesting, which is to explore macromolecular crowding effects that are closer to cellular condition, given that most experiments have been carried out in diluted solutions.

The results are compelling, suggesting that each fold in a fold-switching protein responds differently to crowding effects, such that it can produce shifts in the balance between folds and dictating a preferential stability for one over the other. The authors also delivered control simulations using the single-basin models to confirm the effects of macromolecular crowders.

These results might call the attention of experimentalists to work on the effects of macromolecular crowders on fold-switching, although the GA and GB system might not be the most appropriate one for such endeavor, given that, as the authors mention on page 5, a sequence with a perfect GA and GB population balance has not been reported. Yet, the authors do mention other metamorphic proteins that might be suitable for such studies.

Major Comments: 1) The only major comment that I have is regarding the model in which the tails of GA are impeded to interact with the macromolecular crowders. I am wondering whether the overlap allowed by the removal of the crowding effects in these regions leads to stabilization of the GA fold over the GB fold due to a favorability of a fold with a smaller radius of gyration (here, GA).

Our response: The values of the radius-of-gyration (R_g) obtained for the static G_A and G_B native structures, evaluated over the full chain 1-56 or the middle segment 8-52, are indeed strong indicators of the key result of our manuscript (ie the disordered tails in G_A play a dominant role in the crowding response) because $R_g^{1-56}(G_A) > R_g^{1-56}(G_B)$ while $R_g^{8-52}(G_A) < R_g^{8-52}(G_B)$. However, we believe our “tail” simulations in which regions 1-7 and 53-56 are impeded to interact with the crowders are essential to proving this point. The reason is the significant chain fluctuations (and hence fluctuations in size) that occur in G_A especially, which are not captured by the static R_g values. As you correctly point out, these simulations do indeed indicate that G_A “appears” more compact than G_B from the perspective of the crowders when the tail-crowder interactions are turned off. It is also non-trivial since the tail-crowder interactions are removed regardless of chain conformation.

Given that the disordered regions in the GA fold have different extensions, what would happen if the crowder-protein interactions are systematically deleted for the N-termini and C-termini alone? Are these effects additive?

Our response: Thank you for the nice suggestion. We carried out new simulations in which the segments 1-7 and 53-56 are separately disallowed from interacting with the crowders. The contributions of the tails are indeed approximately linear (see the new Figure S2). We describe these results in two new sentences in section 2.3 on page 8.

Minor Comments:

1) Regarding the dual-basin approach, I am wondering whether the value for the strength of native contacts for the GB fold has anything to do with the proportion of native contacts in the GA and GB folds. Was this value chosen by exploration at random or by looking at the number of native contacts of each fold? Intuitively, the sum of the strength of all native contacts in a given fold should be equal to the sum of the strength of all native contacts in the alternative fold in order to achieve isoenergetic basins, but it could also be that the geometry of the native interactions due to the topology of each fold has an effect on the energetics.

Our response: Indeed, the reason $\kappa^* < 1$ is likely related to the fact that the number native G_B contacts (N_B) is greater than the number of native G_A contacts (N_A). Interestingly, however, the ratio $N_A/N_B = 106/145 = 0.73$ is significantly smaller than $\kappa^* = 0.92$. Hence, the requirement of equal fold populations, does not imply isoenergetic basins. Rather, G_A and G_B rely differently on entropic and energetic factors for stability, as can be seen from their different melting behaviors. See the new Figure S1a and our answer to your minor point 3 below.

2) On the same not, what drove the decision to keep the strength of GA contacts fixed instead of GB?

Our response: There are indeed various ways of varying the relative strength of G_A and G_B contacts. We believe that instead keeping κ_B fixed at unity and varying κ_A , would give similar results. We did not re-produce all results with this alternative setup, but confirmed that a G_A/G_B population balance can be achieved for $\kappa_B = 1$ and $\kappa_A \approx 1.11$ at temperature T_0 (see Figure R1 below). An indication that the physics of the system remains roughly the same is that the ratio $\kappa_B/\kappa_A \approx 0.90$ is comparable to $\kappa^* = 0.92$.

3) The free energy landscapes and populations for each fold are presented at a fixed temperature T in Fig. 1, similar to the total folded populations in Fig. 2. Yet, I wonder whether shifting the temperature to slightly lower or higher values is sufficient to achieve equal populations of each fold and higher values for total folded populations at different strengths of native contacts for the GB fold. Could the authors elaborate whether the plots being presented correspond to the temperature at which the populations of each fold reach its maximum? If not, this would be a reasonable explanation for the minimum P_{tot} value in Fig. 2.

Our response: We have explored the behavior of our model at different temperatures by carrying out additional simulations covering different T s and contact strengths κ_B (still keeping $\kappa_A = 1$ fixed). It is indeed possible to obtain equal fold populations at temperatures different from T_0 . In particular, higher P_{tot} is possible at $T < T_0$. However, in order to maintain $P_A = P_B$, κ^* needs to be adjusted at each T . Based on our additional simulations, we have been able to sketch the relations $\kappa^*(T)$ and $P_{tot}(T)$ (see the new Figure S1b and c). These results are described in two new sentences at the end of section 2.1 on page 5.

The reason κ^* depends on T is likely the distinct melting curves exhibited by G_A and G_B (see Figure S1a). In particular, the melting curves show that G_B dominates over G_A at low enough T , which is likely a consequence of the lower energy of folding for G_B relative to G_A (ultimately deriving from the fact that G_B has a larger number of native contacts than G_A; see point 1 above).

4) *On the same note, it might be possible that the stabilization of GB over GA in the presence of crowders could be due to shifts in the temperature at which both states are under equilibrium. Could the authors produce the plots of the populations of GA and GB as a function of temperature for each crowder packing fraction and for the simulations in the absence of crowders? This would be an indication of stabilization as the authors did for the single-fold systems in Fig. 4, where they checked the increased in T_m due to the presence of crowders.*

Our response: The use of different T_m 's, defined separately for the two folds may be possible but is not as straightforward as quantifying stability using fold populations or free energies. We carried out a set of simulated tempering simulations of G_{AB}^* that allowed us to determine the T -dependence of P_A and P_B across different crowder concentrations (see Figure R2). Although the data is certainly interesting, it is not clear how to analyze these folding curves, especially for G_A . We hope to explore this issue further in future work.

4) *What is the rationale for the choice of crowder particles of radius 12 Å? How is this size comparable to the radius of the beads in the coarse-grained model?*

Our response: Please see our response to Reviewer 2, who raised a similar point.

5) *Which software/simulation package was used for these simulations?*

Our response: Simulations were carried out using an in-house code written in C. We will distribute the code to anyone who request it or, alternatively, publish it under an open-source licence. We will, of course, follow any computer software publishing requirement of the journal.

Figure R1: 2D scatter plot of the root-mean-square deviation (RMSD) taken with respect to either the GA native structure (2KDL) or the GB native structure (2KDM) taken at temperature T_0 .

Figure R2: **Temperature dependence of the crowding effects on GAB* fold-switching.** (a) G_A and (b) G_B fold populations as a function of temperature T for various crowder volume fractions ϕ_c .

REVIEWERS' COMMENTS:

Reviewer #1 (Remarks to the Author):

I'm glad that GFMT helpful to reveal the problem that SPT couldn't, such as the effective radius regarding the tail of repulsive interactions. The problem has been investigated with more expensive/explicit way as in Figure 3 and 4 in J. Phys. Chem. Lett. 2013, 4, 20, 3429–3434. <https://doi.org/10.1021/jz401817x>. Another problem could be revealed is the necessity for volume correction for small simulation box as discussed in ref 35. These effects are small but observable with more sampling. A better way to look at the error is to estimate the error of the mean value for correlated data, <https://github.com/manoharan-lab/flyvbjerg-std-err/>. However, as fitting to model is only a small part of the paper, I will not stress on the things mentioned above. Those are just for information.

A tiny modification,

1. The fitting parameter, δ and R_0 respectively for SPT in Figure 3b, should be present for completion, regardless of goodness.

Reviewer #3 (Remarks to the Author):

The revised version of the manuscript by Bazmi, Seifi and Wallin, presents a computational exploration of the effects of macromolecular crowding on the fold-switch of the GA and GB folds.

In this revised version, the authors made several additional simulations and analyses to address some of the behaviors seen in their simulation systems in the presence of macromolecular crowders, such as ascertaining a proper crowder radius for their analysis, using GFMT for their analysis of macromolecular crowding effects on fold-switching and adding new simulations in which the tails of the GA95 system were independently turned off to ascertain their role in controlling the fold-switch under crowding conditions.

All in all, I believe that this revised version of the manuscript is now suitable for publication. Their results are also timely, given the recent publication of an article that experimentally addresses the effect of crowders in other metamorphic proteins in this journal.

Reviewer #1 (Remarks to the Author):

*I'm glad that GFMT helpful to reveal the problem that SPT couldn't, such as the effective radius regarding the tail of repulsive interactions. The problem has been investigated with more expensive/explicit way as in Figure 3 and 4 in *J. Phys. Chem. Lett.* 2013, 4, 20, 3429–3434. <https://doi.org/10.1021/jz401817x>. Another problem could be revealed is the necessity for volume correction for small simulation box as discussed in ref 35. These effects are small but observable with more sampling. A better way to look at the error is to estimate the error of the mean value for correlated data, <https://github.com/manoharan-lab/flyubjerg-std-err/>.*

However, as fitting to model is only a small part of the paper, I will not stress on the things mentioned above. Those are just for information.

A tiny modification,

1. The fitting parameter, δ and R_0 respectively for SPT in Figure 3b, should be present for completion, regardless of goodness.

Response: We thank the reviewer for bringing our attention to GFMT and for the useful additional information. Regarding the SPT fit parameter δ , and R^0 , these are now provided in a new sentence inserted in section 2.3 on page 8:

“Moreover, the fit to SPT (see Figure 3b), obtained using $R_0 = 13.9 \text{ \AA}$ and giving $\delta = 0.42 \pm 0.01 \text{ \AA}$, is now much better ($\chi^2/(n - 1) \approx 1.5$).”

Reviewer #3 (Remarks to the Author):

The revised version of the manuscript by Bazmi, Seifi and Wallin, presents a computational exploration of the effects of macromolecular crowding on the fold-switch of the GA and GB folds.

In this revised version, the authors made several additional simulations and analyses to

address some of the behaviors seen in their simulation systems in the presence of macromolecular crowders, such as ascertaining a proper crowder radius for their analysis, using GFMT for their analysis of macromolecular crowding effects on fold-switching and adding new simulations in which the tails of the GA95 system were independently turned off to ascertain their role in controlling the fold-switch under crowding conditions.

All in all, I believe that this revised version of the manuscript is now suitable for publication. Their results are also timely, given the recent publication of an article that experimentally addresses the effect of crowders in other metamorphic proteins in this journal.

Response: Thank you.